# GRANULARITY MATTERS IN LONG-TAIL LEARNING

## ABSTRACT

Balancing training on long-tail data distributions remains a long-standing challenge in deep learning. While methods such as re-weighting and re-sampling help alleviate the imbalance issue, limited sample diversity continues to hinder models from learning robust and generalizable feature representations, particularly for tail classes. In contrast to existing methods, we offer a novel perspective on long-tail learning, inspired by an observation: datasets with finer granularity tend to be less affected by data imbalance. In this paper, we investigate this phenomenon through both quantitative and qualitative studies, showing that increased granularity enhances the generalization of learned features in tail categories. Motivated by these findings, we propose a method to increase dataset granularity through category extrapolation. Specifically, we introduce open-set auxiliary classes that are visually similar to existing ones, aiming to enhance representation learning for both head and tail classes. This forms the core contribution and insight of our approach. To automate the curation of auxiliary data, we leverage large language models (LLMs) as knowledge bases to search for auxiliary categories and retrieve relevant images through web crawling. To prevent the overwhelming presence of auxiliary classes from disrupting training, we introduce a neighbor-silencing loss that encourages the model to focus on class discrimination within the target dataset. During inference, the classifier weights for auxiliary categories are masked out, leaving only the target class weights for use. Extensive experiments and ablation studies on three standard long-tail benchmarks demonstrate the effectiveness of our approach, notably outperforming strong baseline methods that use the same amount of data. The code will be made publicly available.

## 1 INTRODUCTION

Deep models have shown extraordinary performance on large-scale curated datasets (He et al., 2016; Simonyan & Zisserman, 2015; Dosovitskiy et al., 2021b). But when dealing with real-world applications, they generally face highly imbalanced (*e.g.*, long-tailed) data distribution: instances are dominated by a few head classes, and most classes only possess a few images (Wang et al., 2021; He et al., 2021; Xiang et al., 2020; Dong et al., 2023). Learning in such an imbalanced setting is challenging as the instance-rich (or head) classes dominate the training procedure (Cui et al., 2021; Samuel & Chechik, 2021; Alshammari et al., 2022; Zhong et al., 2021). Without considering this situation, models tend to classify tailed class samples as similar head categories, leading to significant performance degradation on tail categories (Yu et al., 2022; Park et al., 2022; Parisot et al., 2022; Zhu et al., 2022).

Existing works tackle challenges in long-tail learning from various perspectives. An earlier stream is to re-balance the learning signal (*e.g.*, re-weighting (Cui et al., 2019) and re-sampling (Chawla et al., 2002)). Yet, they inevitably face the scarcity of data and suffer from over-fitting on tail classes (Fig. 1b). Another straightforward fix is to augment training samples into diverse ones through image transformations (DeVries & Taylor, 2017; Zhang et al., 2018; Yun et al., 2019; Chou et al., 2020). These methods typically increase the loss weights or enhance the sample diversity of tail classes to balance representation learning. Despite advances, limited sample diversity still constrains the ability to generalize the learned features. Additionally, improvements in tail class performance are often accompanied by a decline in head class performance. This limitation motivates us to investigate what factors contribute to generalizable feature learning in long-tail settings. Our exploration is inspired by a common, yet counterintuitive, phenomenon observed in existing benchmarks: despite being more imbalanced than ImageNet-LT (Liu et al., 2019), iNat18 (Van Horn et al., 2018)

**Figure 1: Holistic comparison to previous philosophy.** (a) Data imbalance between head and tail classes makes biased features; (b, c): Previous methods are still bounded by existing known classes; (d) We instead seek help from auxiliary open-set data.

achieves nearly balanced performance (see Table 1). This observation raises the question: *Does granularity play a role in the performance balance of long-tail learning?*

To investigate this further, we conducted a pilot study (see Sec. 2.2) using a larger data pool and controlled experiments to verify this phenomenon. We found that datasets with finer granularity are less affected by data imbalance. Feature visualizations (see Fig. 2 and Fig. 9) reveal that, despite a long-tail distribution, datasets with finer granularity enable the model to learn more generalized representations. This discovery motivates us to explore *altering data distribution by introducing open-set categories to increase the granularity of data for long-tail learning.*

| Dataset | #Class | #Train | Granul. | Imb. Ratio $\beta$ | Many | Med. | Few |
|---------|--------|--------|---------|---------------------|------|------|-----|
| IN-LT   | 1000   | 116K   | Coarse  | 5/1280=0.004        | 68.2 | 56.8 | 41.6 |
| iNat18  | 8142   | 438K   | Fine    | 2/1000=0.002        | 70.3 | 71.3 | 70.2 |

**Table 1: Average performance of previous methods.** Results are obtained by averaging the performance listed in Table 3a for ImageNet-LT and Table 3b for iNat18.

At the core of our approach is the idea of augmenting training data with fine-grained categories related to the original ones, thereby increasing granularity (Fig. 1d). To acquire auxiliary data, we establish a fully automated data crawling pipeline powered by the knowledge of large language models (LLMs). Specifically, for each class to be expanded, we query an LLM for $k$ visually similar auxiliary classes, then retrieve corresponding images from the web based on these class names (Fig. 4). The crawled data are subsequently integrated with the original dataset for model training. During training, we introduce a neighbor-silencing loss to enhance discrimination between confusing classes, prevent the model from being overwhelmed by auxiliary classes, and ensure alignment with the objectives of the testing phase. After training, the classifier– by simply masking out the auxiliary classes demonstrates strong performance without the need for additional classifier re-balancing, as required in previous methods (Kang et al., 2020; Zhou et al., 2020).

Intuitively, our method could be interpreted as *category extrapolation*. These augmented categories complete the learning signal, which may fill the gap between originally distinct classes, encourage continuity and smoothness of the feature manifold, and allow better generalization of representations across classes. In terms of classification, samples of auxiliary classes take up the neighborhood of existing classes, thus explicitly enlarging the margin between them and encouraging discriminability. Empirically, we indeed observe tighter clusters and better separation in-between (Fig. 2d).

Our major contributions are summarized as follows:

- We explore the effect of granularity on the performance balance in long-tail learning, which motivate us to introduce neighbor classes to increase the granularity and facilitate representation learning for both head and tail classes.

- We propose a neighbor-silencing learning loss to facilitate long-tail learning with extra open-set categories and design a fully automatic data acquisition pipeline to efficiently harvest data from the Web.

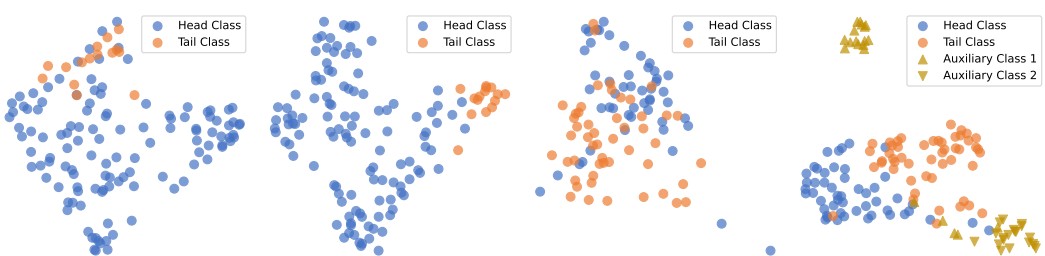

**(a)** Raw feature space (train). **(b)** Baseline after training (train). **(c)** Baseline after training (val). **(d)** After training w/ aux. class (val).

**Figure 2: Feature visualization of confusing head and tail classes by UMAP (McInnes et al., 2020) on ImageNet-LT (Liu et al., 2019).** (a) Raw feature space of training data by DINOv2 (Maxime et al., 2023); (b) Feature space of training data after the training phase; (c) The baseline (re-weighting) shows poor generalization on validation data; (d) Adding auxiliary categories condenses clusters and improves separation.

- We conduct extensive experiments across standard benchmarks using various training paradigms (*e.g.*, random initialization, CLIP (Radford et al., 2021), and DINOv2 (Maxime et al., 2023)), all of which consistently demonstrate high performance. Notably, when training from random initialization, our method improves tail class performance by 16.0% on ImageNet-LT and 8.3% on Places-LT.

## 2 PILOT STUDY

In this section, we investigate whether granularity impacts performance balance in long-tail distribution. We first provide preliminary for long-tail learning and an analysis on a baseline method in Sec. 2.1. Then, we verify the impact of the granularity of training data on long-tail learning (Sec. 2.2) from both quantitative and qualitative perspectives.

### 2.1 PRELIMINARY

In long-tail visual recognition, the model has access to a set of $N$ training samples $\mathcal{S} = \{(\mathbf{x}_n, y_n)\}_{n=1}^{N}$, where $\mathbf{x}_n \in \mathcal{X} \subset \mathbb{R}^D$ and labels $\mathcal{Y} = \{1, 2, .., L\}$. Training class frequencies are defined as $N_y = \sum_{(x_n, y_n) \in \mathcal{S}} \mathbb{1}_{y_n = y}$ and the test-class distribution is assumed to be sampled from a uniform distribution over $\mathcal{Y}$, but is not explicitly provided during training. A classic solution is to minimize the balanced error (BE), of a scorer $\mathbf{f} : \mathcal{X} \to \mathbb{R}^L$, defined as:

$$\mathrm{BE}(\mathbf{x}, \mathbf{f}(\cdot)) = \sum_{y \in \mathcal{Y}} \mathbf{P}_{\mathbf{x}|y}\left( y \notin \arg\max_{y' \in \mathcal{Y}} \mathbf{f}_{y'}(\mathbf{x}) \right), \tag{1}$$

where $\mathbf{f}_y(x)$ is the logit produced for true label $y$ for sample $\mathbf{x}$. Traditionally, this is done by minimizing a proxy loss, the Balanced Softmax Cross Entropy (BalCE) (Cui et al., 2019):

$$\mathcal{L}_{\mathrm{BalCE}}(\mathcal{M}(\mathbf{x}|\theta_f, \theta_w), \mathbf{y}_i) = -\log[p(\mathbf{y}_i|\mathbf{x}; \theta_f, \theta_w)]$$

$$= -\log\left[ \frac{n_{\mathbf{y}_i} e^{z_{\mathbf{y}_i}}}{\sum_{\mathbf{y}_j \in \mathcal{Y}} n_{\mathbf{y}_j} e^{z_{\mathbf{y}_j}}} \right] = \log\left[ 1 + \sum_{\mathbf{y}_j \neq \mathbf{y}_i} e^{\log n_{\mathbf{y}_j} - \log n_{\mathbf{y}_i} + \mathbf{z}_{\mathbf{y}_j} - \mathbf{z}_{\mathbf{y}_i}} \right]. \tag{2}$$

This is known as *re-weighting*, where the contribution of each label's individual loss is scaled by an inverse class frequency derived from the class's instance number $n_{y_i}$. We adopt this setting as the baseline in follow-up experiments.

**On the failure of re-balancing.** The primary challenge of long-tail learning stems from data imbalance, which affects the representation learning of both head classes and few-shot classes. For head classes, if there is a lack of effective negative-class samples, then learning an effective boundary is challenging. To better demonstrate this, we provide feature visualizations of confusing head (Scottish Deerhound) and tail (Irish Wolfhound) classes on ImageNet-LT in **??**. As in Fig. 2a, these two classes are challenging even for the advanced vision foundation model DINOv2 (Maxime et al., 2023). After training with the re-weighting baseline on the imbalanced training data, the learned

features seem relatively satisfactory (Fig. 2b). However, the generalization is poor: samples in the validation data are still convoluted, and the separation between them is unclear (Fig. 2c). On top of this baseline, we then study the effect of data distribution on long-tail learning.

## 2.2 GRANULARITY MATTERS IN LONG-TAIL LEARNING

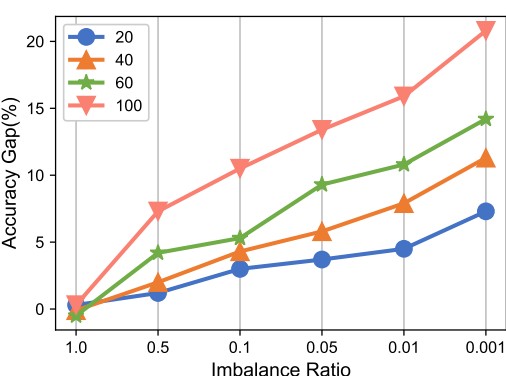

**Figure 3:** Effect of granularity *vs.* imbalance ratio.

We study whether the granularity of the dataset is critical to long-tail learning. Our study is motivated by an intriguing observation that, although more classes and stronger imbalance, we observe nearly balanced performance on iNat18 (Van Horn et al., 2018), as opposed to ImageNet-LT (Liu et al., 2019). A significant distinction is that iNat18 is an extremely fine-grained dataset with over 8000 categories, yet it only consists of 14 superclasses in total. On the other hand, ImageNet-LT, although comprising only 1000 categories, has over 100 superclasses, making it relatively coarse-grained. Therefore, we conduct experiments to study the effect of granularity on long-tail learning.

*Dataset Configuration.* To this end, we construct a dataset pool using ImageNet-21k (Ridnik et al., 2021) and OpenImage (Krasin et al., 2017) datasets. To investigate the influence of granularity, we sample 500 classes from the pool for each time and control the number of superclasses to be $\{20, 40, 60, 100\}$ based on WordNet. Then, we used different imbalance ratios $\{1.0, 0.5, 0.1, 0.05, 0.01, 0.001\}$ to study the effect of granularity on the imbalance ratio. We train the model (ViT-Base (Dosovitskiy et al., 2021b)) using BalCE (Cui et al., 2019) as Eq. (2). We conduct 5 experiments and take the average value.

In Fig. 3, we show the performance gap between head categories and tail categories under different dataset imbalance ratios. The results show that as the granularity increases, the dataset is less sensitive to the imbalance ratio. For example, when the number of superclasses is 20, the performance gap between the head and tail is 7.3%, while the gap is 20.8% when the number of superclasses is 100, under the severe imbalance (imbalance ratio=0.001).

> **Finding 1:** Increased granularity of training data benefits long-tail learning.

In a fine-grained long-tail dataset, although there are few samples for tail categories, many categories share similar patterns, which is conducive to learning distinctive features, thus enhancing generalizability. As reflected in Fig. 2d, for clearer visualization, we sample two fine-grained categories that is denoted as the auxiliary classes. The visualization shows that the separation between head and tail classes is clearly improved. Also, the distribution of intra-class samples is also more compact. Due to the space limitation, we show more examples in Appendix Fig. 9. This motivate us to introduce diverse open-set auxiliary categories to enhance the granularity for close-set long-tail learning.

> **Finding 2:** Despite long-tail distribution, increased granularity could explicitly separate and condensify existing data clusters.

Based on the above findings, given a long-tail dataset, we aim to establish a framework that can effectively acquire auxiliary data to enhance the granularity. Specifically, we utilize LLMs to query the candidate auxiliary categories and crawl images from the Web, followed by a filtering stage to ensure similarity and diversity. To better incorporate auxiliary data for training with target categories, we propose a Neighbor-Silencing Loss to avoid being overwhelmed by auxiliary classes. Details are included in Sec. 3.

# 3 LONG-TAIL LEARNING BY CLASS EXTRAPOLATION

In this section, we first introduce our simple and automatic pipeline for obtaining auxiliary data in Sec. 3.1. Then, we present our new learning objective that effectively leverages the auxiliary data to enhance long-tail learning in Sec. 3.2.

## 3.1 NEIGHBOR CATEGORY SEARCHING

In search of neighbor categories sharing some common visual patterns with the pre-defined categories in the dataset, we design a fully automatic crawling pipeline that includes (i) querying neighbor categories from LLMs to obtain similar categories and enhance the granularity of the training data and (ii) retrieving corresponding images from the web and conducting filtering to guarantee similarity and diversity. An overview of this pipeline is illustrated in Fig. 4, and we introduce each step in detail as follows.

**Querying LLM for Neighbor Categories.** We take advantage of the recent development of Large Language Models (LLMs), *e.g.*, GPT-4 (OpenAI, 2023), and query them for expert knowledge of possible visually similar classes with respect to the classes to extrapolate (*i.e.*, the medium and tail classes by default). For example, we can prompt the language model with: "Please create a list which contains 5 visually similar categories of {CLS}". However, the output of this naive prompt is unstable, possibly because 'visually similar categories' by itself is quite a broad and vague concept. To make the prompt more concrete and clear for LLMs, we design a structural prompt with in-context learning:

> **Task:** Given a category name, please list out 5 classes that are visually similar to the provided classes.
> **Query:** sports car
> **Response:** sedan, coupe, SUV, luxury car, electric car
> **Query:** {CLS}
> **Response:**

The LLM then completes the response above. After that, classes in the target dataset $\mathcal{S}$ are filtered out to avoid possible information leaks. Then, the remaining class names are fed to an image-searching engine for image retrieval.

**Retrieving and Filtering Images from the Web.** Images retrieved by the search engine can be noisy, thus, a filtering strategy is adopted. An image $\mathbf{x}_r$ corresponding to a specific class $y_i$ is dropped if: (i) the class's name does not exist in the associated caption; or (ii) the visual similarity between the class and this image satisfies thresholds: $\gamma_1 < \cos(\mathbf{p}_i, \mathbf{f}_r) < \gamma_2$. We employ DINOv2 (Maxime et al., 2023) for feature extraction and use cosine similarity as the metric. Specifically, the prototype $\mathbf{p}_i$ of category $y_i$ is computed as the average feature of all samples of this category in the target dataset $\mathcal{S}$: $\mathbf{p}_i = {}^1/n_{y_i} \sum_j \mathbf{f}_j$. After the filtering pro-

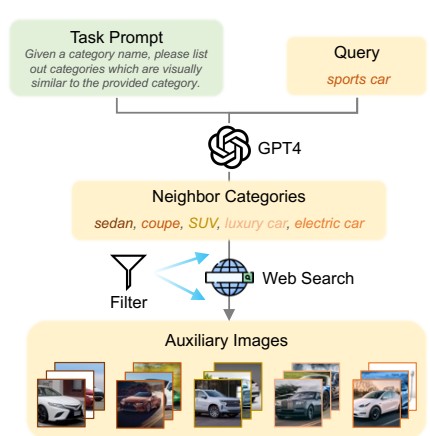

**Figure 4: Data crawling pipeline.** We prompt GPT-4 for visual-similar categories of query classes and retrieve corresponding images from the web. Classes already in the label set and images of lower visual similarity than the threshold are filtered out.

cess, the model has access to a set of $M$ auxiliary training samples $\mathcal{A} = \{(\mathbf{x}_m, y_m)\}_{m=1}^M$, where $\mathbf{x}_m \in \mathcal{X} \subset \mathbb{R}^D$ and labels $\mathcal{Y}_a = \{L+1, L+2, .., L+K\}$ and the category number for auxiliary set is $K$.

## 3.2 Learning with Auxiliary Categories

We mix the auxiliary dataset $\mathcal{A}$ and the target dataset $\mathcal{S}$ for training. A naive approach is to directly employ the BalCE loss (Cui et al., 2019) by merging the label space:

$$\mathcal{L}_{\text{BalCE}} = -\log\left[n_{\mathbf{y}_i} e^{z_{\mathbf{y}_i}} / (\overbrace{\sum_{\mathbf{y}_j \in \mathcal{Y}} n_{\mathbf{y}_j} e^{z_{\mathbf{y}_j}}}^{\text{Target}} + \overbrace{\sum_{\mathbf{y}_j \in \mathcal{Y}_a} n_{\mathbf{y}_j} e^{z_{\mathbf{y}_j}}}^{\text{Auxiliary}})\right]. \tag{3}$$

But note that our objective is to classify $L$ categories within the target dataset, as opposed to $L + K$ categories. Directly employing the standard BalCE loss as Eq. (3) would result in an inconsistency between the optimization process and the ultimate goal. The auxiliary part could overwhelm optimization and result in degenerated performance. We thus "silent" them by weighting as follows.

**Silencing the Overwhelming Neighbors.** Concretely, if $y_j$ is a neighbor category of $y_i$ from auxiliary categories, we spot this as possible neighbor overwhelming and give the corresponding logit a smaller weight. To clarify, $y_j$ is a neighbor category of $y_i$ means that $y_j$ is queried from $y_i$ by Neighbor Category Searching (Sec. 3.1). We thus expect the auxiliary classes to influence less the target class which they are queried from, and contribute more to their classification as a whole with respect to other classes. The neighbor-silencing variant of the re-balancing loss is then formulated as:

$$\mathcal{L}_{\text{NS-CE}} = \log\left[1 + \sum_{\mathbf{y}_j \neq \mathbf{y}_i} \lambda_{ij} \cdot e^{\log n_{\mathbf{y}_j} - \log n_{\mathbf{y}_i} + \mathbf{z}_{\mathbf{y}_j} - \mathbf{z}_{\mathbf{y}_i}}\right], \tag{4}$$

where $\lambda_{ij} = \lambda_s$, if $y_i$ and $y_j$ satisfy that one is the other's neighbor category and one of them from auxiliary categories, and $\lambda_{ij} = 1$ otherwise. $\lambda_s$ is the weight for balancing the loss between neighbor category pairs and non-pairs. By default, $0 < \lambda_s < 1$. In this way, we assign a smaller weight to neighbor category pairs, thus, the effect within neighbor classes is weakened, and the optimization focuses more on their separation as a whole from other confusing classes.

**Obtaining the Final Classifier.** Given that our model's classifier includes more categories, it cannot be directly applied to the target dataset for evaluation. A common practice is to discard the trained classifier and re-train it with re-balancing techniques on the target dataset through linear probing (Kang et al., 2020; Zhou et al., 2020). However, this could be suboptimal since the separation hyper-planes shaped by auxiliary categories can be undermined. Therefore, we try directly masking out the weights of auxiliary categories, retaining only the weights of the target categories. Specifically, we denote the trained classifier weights as $\mathcal{W} = \{\mathbf{w}_i\}_{i=1}^{L+K}$, where $\mathbf{w}_i \subset \mathbb{R}^C$, and keep $\mathcal{W} = \{\mathbf{w}_i\}_{i=1}^{L}$. Surprisingly, this simpler approach works better. This is potentially because incorporating more auxiliary fine-grained categories can enable the classifier to focus on class-specific discriminative features. These features possess stronger generalizability, facilitating the classifier to construct more precise separation hyper-planes.

## 4 Experiments

In this section, we first introduce benchmark datasets for long-tail image classification in Sec. 4.1, followed by the implementation details in Sec. 4.2. Then, we compare our approach with the baseline and state-of-the-art methods in Sec. 4.3. Finally, a series of ablation studies are performed for further analysis in Sec. 4.4.

### 4.1 Datasets

We experiment with three standard long-tailed image classification benchmarks. All datasets adopt the official validation/test images for fair comparisons. We report accuracy on three splits of the set of classes: Many-shot (more than 100 images), Medium-shot (20~100 images), and Few-shot (less than 20 images). Besides, we also report the commonly used top-1 accuracy over all classes for evaluation. Detailed dataset information is available in the Appendix.

**Table 2: Quantitative results of the proposed method on three standard benchmarks.** For each dataset, we conduct three pre-training paradigms (training from scratch, CLIP, and DINOv2) to compare our method with baseline methods on accuracy (%). In addition, we report the relative improvement of our method compared to the baseline method in each setting.

| | Method | ImageNet-LT | | | | iNaturalist 18 | | | | Place-LT | | | |
|---|---|---|---|---|---|---|---|---|---|---|---|---|---|
| | | Overall | Many | Med. | Few | Overall | Many | Med. | Few | Overall | Many | Med. | Few |
| Scratch | Baseline | 60.9 | 72.9 | 56.8 | 41.4 | 76.1 | 78.5 | 76.9 | 74.6 | 39.9 | 43.0 | 40.5 | 33.3 |
| | + *Ours* | 68.2$_{\uparrow 7.3}$ | 74.5$_{\uparrow 1.6}$ | 66.2$_{\uparrow 9.4}$ | 57.4$_{\uparrow 16.0}$ | 78.0$_{\uparrow 1.9}$ | 78.9$_{\uparrow 0.4}$ | 78.2$_{\uparrow 1.3}$ | 77.5$_{\uparrow 2.9}$ | 43.8$_{\uparrow 3.9}$ | 43.7$_{\uparrow 0.7}$ | 44.8$_{\uparrow 4.3}$ | 41.6$_{\uparrow 8.3}$ |
| CLIP | Baseline | 74.0 | 77.2 | 72.8 | 68.5 | 75.0 | 77.8 | 76.5 | 72.5 | 48.4 | 47.9 | 48.6 | 48.9 |
| | + *Ours* | 77.3$_{\uparrow 3.5}$ | 79.1$_{\uparrow 1.9}$ | 76.8$_{\uparrow 4.0}$ | 74.1$_{\uparrow 5.6}$ | 78.5$_{\uparrow 3.5}$ | 79.5$_{\uparrow 1.5}$ | 79.3$_{\uparrow 2.8}$ | 77.3$_{\uparrow 4.8}$ | 50.5$_{\uparrow 2.1}$ | 50.0$_{\uparrow 2.1}$ | 51.0$_{\uparrow 2.4}$ | 50.2$_{\uparrow 1.3}$ |
| DINOv2 | Baseline | 79.6 | 84.3 | 78.3 | 71.1 | 85.0 | 85.7 | 86.2 | 84.2 | 49.5 | 49.2 | 51.3 | 46.1 |
| | + *Ours* | 82.0$_{\uparrow 2.4}$ | 84.7$_{\uparrow 0.4}$ | 81.5$_{\uparrow 3.2}$ | 76.2$_{\uparrow 5.1}$ | 87.0$_{\uparrow 2.0}$ | 86.4$_{\uparrow 0.7}$ | 87.4$_{\uparrow 1.2}$ | 86.7$_{\uparrow 2.5}$ | 50.8$_{\uparrow 1.3}$ | 49.4$_{\uparrow 0.2}$ | 52.4$_{\uparrow 1.1}$ | 49.2$_{\uparrow 3.1}$ |

**ImageNet-LT** (Liu et al., 2019) is a class-imbalanced subset of the popular image classification benchmark ImageNet ILSVRC 2012 (Russakovsky et al., 2015). The images are sampled following the *Pareto* distribution with a power value $\alpha = 6$, containing 115.8k images from 1,000 categories. **iNaturalist 2018** (Van Horn et al., 2018) (iNat18 for short) is a species classification dataset, which consists of 437.5k images from 8,142 fine-grained categories following an extreme long-tail distribution. **Places-LT** is a synthetic long-tail variant of the large-scale scene classification dataset Places (Zhou et al., 2017). With 62.5k images from 365 categories, its class cardinality ranges from 5 to 4,980.

## 4.2 IMPLEMENTATION DETAILS

We adopt ViT-Base (Dosovitskiy et al., 2021b) as the backbone. Our models are trained with the AdamW optimizer (Loshchilov & Hutter, 2019) with $\beta_s = \{0.9, 0.95\}$, with an effective batch size of 512 on 4 NVIDIA 3090 GPUs. We train all models with $\mathrm{RandAug}(9, 0.5)$ (Cubuk et al., 2020), $\mathrm{Mixup}(0.8)$ (Zhang et al., 2018) and $\mathrm{Cutmix}(1.0)$ (Yun et al., 2019). We set the maximum sampling number for each auxiliary category to 50 in each training epoch. For the ratio of neighbor category for head, medium, and tail class, we set to $1 : \left\lceil \frac{N_h}{N_m} \right\rceil : \left\lceil \frac{N_h}{N_t} \right\rceil$, where $N_h$, $N_m$, and $N_t$ denote the total number of samples of head, medium, and tail classes, respectively. $\lceil \cdot \rceil$ stands for ceiling, which rounds a number up to the nearest integer. Following LiVT (Xu et al., 2023), the training epochs for ImageNet-LT, iNaturalist, and Place-LT is set to 100, 100, and 30, respectively. The hyper-parameter $\lambda_s$ is set to 0.1. See detailed implementation settings in the Appendix.

## 4.3 MAIN RESULTS

**Comparison with Baseline with Different Pre-training.** We experiment with three different pre-training paradigms (*i.e.*, random initialization, CLIP (Radford et al., 2021), and DINOv2 (Maxime et al., 2023)). The baseline applies BalCE (Cui et al., 2019) loss. As shown in Table 2, our method significantly improves the performance over the baseline on all three datasets, especially on fewer-shot classes. This improvement is also consistent and generalizes to a variety of pre-training strategies. In particular, when the model is trained from scratch, we observe a significant performance boost on ImageNet-LT, with a 16.0% increase in accuracy on the tail classes. A plausible explanation is that randomly initialized networks are more prone to overfitting on tail classes compared to large-scale pre-trained models. Our method effectively addresses this issue by utilizing neighbor categories. Besides, even with pre-trained models as initialization, our approach consistently demonstrates satisfactory improvements. For example, when using DINOv2 as the backbone, we achieve performance improvements of 5.0%, 2.5%, and 3.1% on the tail classes of ImageNet-LT, iNaturalist, and PlaceLT datasets, respectively, without compromising performance on the head classes. This verifies our method's generalizability and effectiveness on long-tail datasets.

**Can Learning by Class Extrapolation Enhance the State-of-the-Art Methods?** We conduct comprehensive experiments with existing SoTAs in Table 3a, Table 3b, and Table 4. Current methods can be generally categorized into two settings, *i.e.*, training from scratch or adopting CLIP

**Table 3:** Performance on ImageNet-LT and iNaturalist 2018. We report accuracy (%) of all methods under three pre-training paradigms (*indicates using additional text information and related modules for training. For each pre-training paradigm, we select a SOTA method, and add proposed method with the auxiliary data on it. We also report the performance of adding the auxiliary data but without our method, which denotes by †.)

**(a) Performance on ImageNet-LT.**

| Methods | Backbone | Overall | Many | Med. | Few |
|---|---|---|---|---|---|
| **Training from scratch** | | | | | |
| MiSLAS (Zhong et al., 2021) | ResNet-50 | 52.7 | 62.9 | 50.7 | 34.3 |
| RIDE (Wang et al., 2021) | ResNet-50 | 56.8 | 68.2 | 53.8 | 36.0 |
| LA (Menon et al., 2021) | ResNet-50 | 51.1 | - | - | - |
| DisAlign (Zhang et al., 2021) | ResNet-50 | 52.9 | 61.3 | 52.2 | 31.4 |
| BCL (Zhu et al., 2022) | ResNet-50 | 56.0 | - | - | - |
| PaCo (Cui et al., 2021) | ResNet-50 | 57.0 | - | - | - |
| NCL (Li et al., 2022) | ResNet-50 | 57.4 | - | - | - |
| LiVT (Xu et al., 2023) | ViT-B | 60.9 | 73.6 | 56.4 | 41.0 |
| LiVT† (Xu et al., 2023) | ViT-B | 59.3 | 74.2 | 54.1 | 35.3 |
| *Ours* | ViT-B | **68.2** | **74.5** | **66.2** | **57.4** |
| **Fine-tuning pre-trained model (CLIP)** | | | | | |
| BALLAD (Ma et al., 2021) | ViT-B | 75.7 | 79.1 | 74.5 | 69.8 |
| VL-LTR* (Tian et al., 2022) | ViT-B | 77.2 | **84.5** | 74.6 | 59.3 |
| Decoder (Wang et al., 2023) | ViT-B | 73.2 | - | - | - |
| LIFT (Shi et al., 2024) | ViT-B | 77.0 | 80.2 | 76.1 | 71.5 |
| LIFT† (Shi et al., 2024) | ViT-B | 75.4 | 80.3 | 73.8 | 67.1 |
| *Ours* | ViT-B | **78.8** | 80.3 | **78.4** | **75.8** |
| **Fine-tuning pre-trained model (DINOv2)** | | | | | |
| LiVT (Xu et al., 2023) | ViT-B | 79.6 | 84.3 | 78.3 | 71.1 |
| LiVT† (Xu et al., 2023) | ViT-B | 77.9 | 84.4 | 75.6 | 67.8 |
| *Ours* | ViT-B | **82.0** | **84.7** | **81.5** | **76.2** |

**(b) Performance on iNaturalist 2018.**

| Method | Backbone | Overall | Many | Med. | Few |
|---|---|---|---|---|---|
| **Training from scratch** | | | | | |
| cRT (Kang et al., 2020) | ResNet-50 | 65.2 | 69.0 | 66.0 | 63.2 |
| MiSLAS (Zhong et al., 2021) | ResNet-50 | 71.6 | 73.2 | 72.4 | 70.4 |
| DiVE (He et al., 2021) | ResNet-50 | 69.1 | 70.6 | 70.0 | 67.6 |
| DisAlign (Zhang et al., 2021) | ResNet-50 | 69.5 | 61.6 | 70.8 | 69.9 |
| BCL (Zhu et al., 2022) | ResNet-50 | 71.8 | - | - | - |
| PaCo (Cui et al., 2021) | ResNet-50 | 73.2 | 70.4 | 72.8 | 73.6 |
| NCL (Li et al., 2022) | ResNet-50 | 74.2 | 72.0 | 74.9 | 73.8 |
| GML (Suh & Seo, 2023) | ResNet-50 | 74.5 | - | - | - |
| LiVT (Xu et al., 2023) | ViT-B | 76.1 | 78.9 | 76.5 | 74.8 |
| LiVT† (Xu et al., 2023) | ViT-B | 66.2 | 78.0 | 68.2 | 60.4 |
| *Ours* | ViT-B | **78.0** | **78.9** | **78.2** | **77.5** |
| **Fine-tuning pre-trained model (CLIP)** | | | | | |
| VL-LTR* (Tian et al., 2022) | ViT-B | 76.8 | - | - | - |
| Decoder (Wang et al., 2023) | ViT-B | 59.2 | - | - | - |
| LIFT (Shi et al., 2024) | ViT-B | 79.1 | 72.4 | 79.0 | 81.1 |
| LIFT† (Shi et al., 2024) | ViT-B | 74.5 | 72.9 | 75.3 | 73.9 |
| *Ours* | ViT-B | **80.9** | **79.6** | **80.1** | **82.1** |
| **Fine-tuning pre-trained model (DINOv2)** | | | | | |
| LiVT (Xu et al., 2023) | ViT-B | 85.0 | 85.7 | 86.2 | 84.2 |
| LiVT† (Xu et al., 2023) | ViT-B | 82.9 | 85.9 | 84.1 | 80.4 |
| *Ours* | ViT-B | **87.0** | **86.4** | **87.4** | **86.7** |

pre-training. For a fair comparison, we benchmark our method regarding each setting correspondingly. Under the train-from-scratch setting, we implement our method based on LiVT. The results show that our approach outperforms alternative methods by a significant margin. Specifically, when compared to LiVT (Xu et al., 2023), we observe improvements of 16.4%, 2.7%, and 14.1% in the tail classes across the three datasets. When CLIP pre-training is adopted, our method still achieves the best performance. Under the CLIP pre-training setting, we implement our method based on LIFT (Shi et al., 2024). Notably, we do not introduce additional complex structures as in VL-LTR (Tian et al., 2022). Besides, we also present results obtained by DINOv2, in which we provide the results of LiVT initialized by pre-trained weights from DINOv2. In this setting, our method also shows considerable improvements.

**Fair Comparison.** In each pre-training paradigm (Table 3a, Table 3b, and Table 4), we select a SOTA method, and add proposed method with the auxiliary data on it, which is denoted by †. When using the neighbor categories with other methods, we can observe that the performance in medium and few classes declines. The potential reason is that the representation learning of medium and few classes are overwhelmed by auxiliary categories, which indicates the effectiveness of our proposed methods.

**Comparison with methods fine-tuned with extra data.** As shown in Table 5, we compare our methods with approaches trained with extra data. Note that VL-LTR (Tian et al., 2022) collects textual descriptions for each category as auxiliary data. RAC (Long et al., 2022) retrieves samples in a data pool with 11.2M images and leverages the most similar samples to refine features during inference. Our method only utilizes 3.6M auxiliary images and surpasses them by a large margin.

## 4.4 ABLATION AND ANALYSIS

**Contributions of Individual Components.** As shown in Tab. 6, we evaluate the contribution of each component of the full method. The baseline is BalCE with DINOv2 pretraining. We conduct ablation experiments on ImageNet-LT. We replace the re-balancing loss (Eq. (2)) with the neighbor-silencing loss (Eq. (4)), obtaining improvements of 1.0% and 1.9% in the medium and tail categories, respectively. If we use the direct classifier instead of retraining the classifier by linear probing, the performance in the medium and tail categories increases to 79.2% and 73.2%, respectively. The best performance is achieved when we do not re-train the classifier and instead directly utilize the classifier weights corresponding to the target categories.

**Table 4: Performance on Places-LT.**

| Method | Backbone | Overall | Many | Med. | Few |
|---|---|---|---|---|---|
| **Training from scratch** | | | | | |
| MiSLAS (Zhong et al., 2021) | ResNet-152 | 40.4 | 39.6 | 43.3 | 36.1 |
| DisAlign (Zhang et al., 2021) | ResNet-152 | 39.3 | 40.4 | 42.4 | 30.1 |
| ALA (Zhao et al., 2022) | ResNet-152 | 40.1 | 43.9 | 40.1 | 32.9 |
| PaCo (Cui et al., 2021) | ResNet-152 | 41.2 | 36.1 | 47.9 | 35.3 |
| LiVT (Xu et al., 2023) | ViT-B | 40.8 | **48.1** | 40.6 | 27.5 |
| LiVT† (Xu et al., 2023) | ViT-B | 39.1 | 48.2 | 37.8 | 25.3 |
| *Ours* | ViT-B | **43.8** | 43.7 | **44.8** | **41.6** |
| **Fine-tuning pre-trained model (CLIP)** | | | | | |
| BALLAD (Ma et al., 2021) | ViT-B | 49.5 | 49.3 | 50.2 | 48.4 |
| VL-LTR* (Tian et al., 2022) | ViT-B | 50.1 | **54.2** | 48.5 | 42.0 |
| Decoder (Wang et al., 2023) | ViT-B | 46.8 | - | - | - |
| LIFT (Shi et al., 2024) | ViT-B | 51.5 | 51.3 | 52.2 | 50.5 |
| LIFT† (Shi et al., 2024) | ViT-B | 48.8 | 51.5 | 48.3 | 45.1 |
| *Ours* | ViT-B | **50.5** | 50.0 | **51.0** | 50.2 |
| **Fine-tuning pre-trained model (DINOv2)** | | | | | |
| LiVT (Xu et al., 2023) | ViT-B | 49.5 | 49.2 | 51.3 | 46.1 |
| LiVT† (Xu et al., 2023) | ViT-B | 46.8 | 49.0 | 46.8 | 42.9 |
| *Ours* | ViT-B | **52.4** | **51.6** | **53.0** | **52.3** |

**Table 5: Comparison with methods fine-tuned with extra data.** Our results are notably stronger despite better sample efficiency.

| Method | #Data | Overall | Many | Med. | Few |
|---|---|---|---|---|---|
| **Results on iNat18 with ViT-B as backbone** | | | | | |
| VL-LTR (Tian et al., 2022) | Texts | 76.8 | - | - | - |
| RAC (Long et al., 2022) | 11.2M | 80.2 | 75.9 | 80.5 | 81.1 |
| *Ours* | 3.6M | **87.0** | **86.4** | **87.4** | **86.7** |

**Table 6: Contributions of individual components.** Results are obtained on ImageNet-LT.

| Method | Many | Med. | Few | Overall |
|---|---|---|---|---|
| Baseline | 84.3 | 78.3 | 71.1 | 79.6 |
| + Neighbor Silencing | 84.5 | 80.8 | 75.5 | 81.5 |
| + Direct Classifier | 84.4 | 79.2 | 73.2 | 80.4 |
| + both | 84.7 | 81.5 | 76.2 | 82.0 |

The curation of the auxiliary dataset primarily involves three hyper-parameters: the number of auxiliary categories associated with a target category, the maximum number of samples per auxiliary class, and the proportion of the number of auxiliary categories for head ($aux_{head}$), medium ($aux_{medium}$), and tail classes ($aux_{tail}$), i.e. $aux_{head} : aux_{medium} : aux_{tail}$ (denoted as auxiliary sampling ratio for simplicity). We will analyze these three hyper-parameters separately and fix the other two hyper-parameters individually. The default values for these three hyper-parameters are 5, 50, and 1:1:3, respectively.

**Number of Sampled Categories.** Fig. 6a studies the effect of the number of auxiliary categories for each target class. The optional values are set to $\{1, 3, 5, 7, 8\}$. We can observe that as the number of neighbor categories increases, the performance gradually improves and finally saturates when approaching 5.

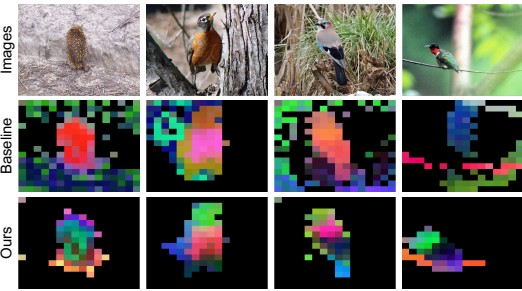

**Figure 5: PCA visualization of "Tail" images in ImageNet-LT.** Top-3 PCA components of features are mapped to RGB channels.

**Maximum Number of Sampled Instances Per Class.** As shown in Fig. 6b, we study the effect of the number of samples per neighbor category. The optional values are $\{10, 30, 50, 100, 150\}$. If the number of samples collected for a class exceeds the limit, we randomly subsample it to the corresponding number; and if less, we keep them unchanged. It can be seen that as the limit increases to 50, the performance improves. However, when too many instances are included, the performance drops. This can be attributed to an excessive number of samples from auxiliary classes, resulting in an overwhelming of these categories.

**Auxiliary Sampling Ratio.** Fig. 6c studies the proportion of the number of auxiliary categories for head, medium, and tail classes. When the ratio is 0:1:3, which indicates that the neighbor categories for many classes are removed, we can observe a performance degradation in many classes from 84.4% to 82.3%. This could be because, with only the addition of auxiliary data in the medium and few-shot categories, feature learning tends to skew towards these medium and few-shot categories. Moreover, when we decrease the ratio on medium (ratio=1:0.5:3) and tail (ratio=1:1:1) classes, the performance degrades, respectively.

**Visualization.** Fig. 5 shows the top-3 PCA components of images sampled from "Tail" classes of ImageNet-LT, where each component is mapped to an RGB channel, and the background is removed by thresholding the first PCA component. Both the baseline (Cui et al., 2019) and our method adopt DINOv2 pre-training. While the baseline finds it hard to locate the object of interest, our method clearly captures better objectness despite the scarcity of "Tail" images.

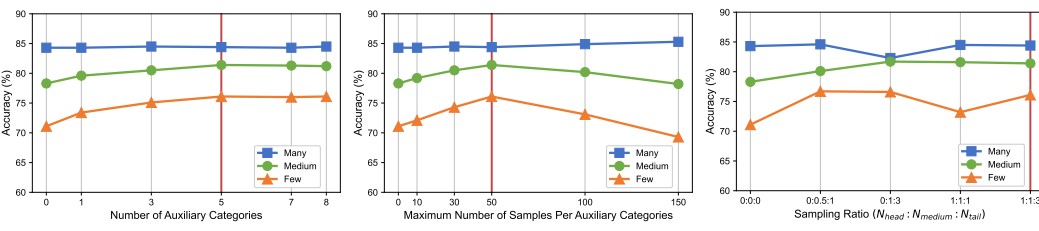

**(a)** Effect of #aux. categories.     **(b)** Effect of #sample/aux. class.     **(c)** Effect of sampling ratio.

Figure 6: **Ablation study on factors related to the curation of auxiliary dataset.** Experiments are conducted on ImageNet-LT (Liu et al., 2019). Default options are marked in red.

## 5   RELATED WORKS

**Re-Balancing Long-Tail Learning.**     Class-level re-balancing methods include oversampling training samples from tail classes (Chawla et al., 2002), under-sampling data points from head classes (Liu et al., 2006), and re-weighting the loss values or gradients based on label frequencies (Cao et al., 2019; Cui et al., 2019) or model's predictions (Lin et al., 2017). Classifier re-balancing mechanisms are based on the finding that uniform sampling on the whole dataset during training benefits representation learning but leads to a biased classifier, so they design specific algorithms to adjust the classifier during or after the representation learning phase (Zhou et al., 2020; Kang et al., 2020).

**Data Augmentation for Long-Tail Learning.**   Spatial augmentation methods have performed satisfactorily for representation learning. Among these approaches, Cutout (DeVries & Taylor, 2017) removes random regions, CutMix (Yun et al., 2019) fills the removed regions with patches from other images, and Mixup series (Zhang et al., 2018; Verma et al., 2019; Summers & Dinneen, 2019) performs convex combination between images. Since data augmentation is closely related to over-sampling, it is also adopted by recent long-tail recognition literature (Zhou et al., 2020; Zhong et al., 2021). These techniques, however, are adopted directly while overlooking special data distributions in long-tail learning. Recently, Remix (Chou et al., 2020) was proposed in favor of the minority classes when mixing samples. Yet, this is still bounded by existing classes. Unlike above, our method samples images from open-set distributions and could greatly benefit from higher data diversity.

**Auxiliary Resources for Long-Tail Learning.**   Previous efforts mainly lie in refining representations with fixed external image features encoded by pre-trained models (Long et al., 2022; Iscen et al., 2023). The external data could be either the training dataset (Long et al., 2022) or crawled from the web (Iscen et al., 2023), and the fusing process could be either non-parametric (Long et al., 2022) or learned in an attentive fashion (Iscen et al., 2023). Besides images, another line (Tian et al., 2022) is to leverage external textual descriptors encoded by vision-language models (Radford et al., 2021). Our method, instead, poses a clear contrast by explicitly introducing external open-set data into a clean training pipeline and is not dependent on any foundation model. There is also a recent work in self-supervised learning that shares the idea of crawling visually-similar data for task-specific improvements (Li et al., 2023). Instead, our work places a special focus on long-tail learning.

## 6   CONCLUDING REMARKS

This paper introduces category extrapolation, which leverages diverse open-set images crawled from the web to enhance closed-set long-tail learning. In addition to a clean and decent method that shows superior performance on "Medium" and "Few" splits across standard benchmarks, we also provide instrumental guidance on when the auxiliary data helps most and empirical explanations on how they help shape the feature manifold through visualizations. We hope our research will attract more researchers to consider how to leverage additional data to address the pervasive problem in long-tail learning. Related research topics could include (i) what kind of additional data is more compatible with target datasets and (ii) how to take the additional data in conjunction with target datasets for training.

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

# A APPENDIX

In this supplementary material, we first provide more implementation details in Appendix B about training configurations (Appendix B.1) and auxiliary data collection (Appendix B.2). Then we conduct additional experiments in Appendix C including an experimental comparison to improved SOTA with DIONOv2 (Appendix C.1), and extended ablation studies (Appendix C.2) related to $\lambda_s$ in the proposed neighbor-silencing loss and the number of samples in the auxiliary dataset, and feature visualization to validate the effectiveness of auxiliary categories (Appendix C.3), and analysis for long-tail in iNaturalist18 (Van Horn et al., 2018) (Appendix C.4). In Appendix D, we discuss our contributions (Appendix D.1), limitations (Appendix D.2), and future work (Appendix D.3).

# B IMPLEMENTATION DETAILS

## B.1 TRAINING

We employ LiVT (Xu et al., 2023) as our baseline since it achieves the top performance under the training from scratch paradigm using ViT (Dosovitskiy et al., 2021a). Specifically, when training from scratch, following LiVT (Xu et al., 2023), we conduct MAE (He et al., 2022) training on the downstream dataset because training directly on a long-tail dataset with randomly initialized parameters makes it difficult to converge. When using pre-training paradigms of CLIP and DINOv2, we directly initialize ViT from their weights. Furthermore, the models are trained with AdamW optimizer (Loshchilov & Hutter, 2019) with $\beta_s = \{0.9, 0.95\}$, with an effective batch size of 512 on 4 NVIDIA 3090 GPUs. The values for weight decay and layer decay are 0.05 and 0.75, respectively. We train all models with $\mathrm{RandAug}(9, 0.5)$ (Cubuk et al., 2020), $\mathrm{Mixup}(0.8)$ (Zhang et al., 2018) and $\mathrm{Cutmix}(1.0)$ (Yun et al., 2019). Following LiVT (Xu et al., 2023), the number of training epochs for ImageNet-LT, iNaturalist 18, and Place-LT is set to 100, 100, and 30, respectively. The number of epochs for warmup is set to 10, 10, and 5. The learning rate is set to 1e-3, 1e-5, and 3.5e-5 for training from scratch, CLIP, and DINOv2, respectively. We set a cosine learning rate schedule and the minimum learning rate is 1e-6. We set the maximum sampling number for each auxiliary category to 50 in each training epoch. The hyper-parameter $\lambda_s$ is set to 0.1. For the ratio of neighbor category for head, medium, and tail classes, we set to $1 : \left\lceil \frac{N_h}{N_m} \right\rceil : \left\lceil \frac{N_h}{N_t} \right\rceil$, where $N_h$, $N_m$, and $N_t$ denote the instance number of head, medium, and tail classes, respectively. $\lceil \cdot \rceil$ stands for ceiling, which rounds a number up to the nearest integer.

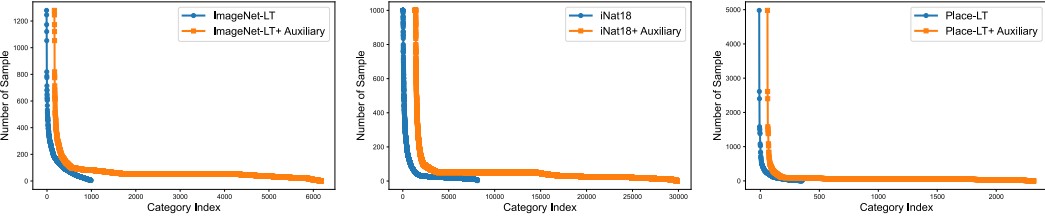

**Figure 7: Distribution of samples of original datasets and corresponding datasets with auxiliary data.** Please note that because two lines partially overlap, for a better display, the index of the augmented dataset is slightly shifted.

## B.2 DATA COLLECTION

We leverage GPT-3.5/4 (OpenAI, 2023) to search names of visually similar categories for the downstream long-tail datasets. We design a structural prompt with in-context learn-

**Table 7: Examples of query classes and respective auxiliary classes across three datasets.**

| Query | Neighbor Categories |
|---|---|
| **ImageNet-LT** | |
| Wolf Spider | Grass Spider, Fishing Spider, Funnel Web Spider, Garden Spider, Dock Spider, huntsman spider |
| Irish Wolfhound | Greyhound, Pharaoh hound, Silken Windhound, Coonhound, Plott Hound, Bearded Collie |
| Basketball | Handball, Football, Badminton Shuttlecock, Softball, Cricket Ball, Billiard Ball, Bowling Ball |
| Kingsnake | Milk Snake, Corn Snake, Hognose Snake, Ribbon Snak, Black Racer, Speckled Kingsnake |
| **iNaturalist 18** | |
| Dryopteris Expansa | Dryopteris Austriaca, Dryopteris Carthusiana, Dryopteris Dilatata, Dryopteris Filixmas |
| Polypodium Virginianum | Polypodium Amorphum, Polypodium Californicum, Polypodium Vulgare, Polypodium Scouleri |
| Adiantum Hispidulum | Adiantum Diaphanum, Adiantum Raddianum, Adiantum Reniforme, Adiantum Venustum |
| Spilosoma Lubricipeda | Arctia Caja, Arctia Villica, Callimorpha Dominula, Diaphora Mendica, Eilema Depressa |
| **Place-LT** | |
| Bus Interior | Airplane Interior, Tram Interior, Subway Interior, Van Interior, Taxi Interior, Limo Interior |
| Bamboo Forest | Tropical forest, Evergreen Forest, Pine Forest, Birch Forest, Cypress Forest, Mangrove Forest |
| Fastfood Restaurant | Seafood Restaurant, Vegetarian Restaurant, Pizza Restaurant, Mexican Restaurant, Steakhouse |
| Physics Laboratory | Materials Laboratory, Environmental Laboratory, Geology Laboratory, Engineering Laboratory |

ing and the below shows one example of our interaction with GPT-4 (OpenAI, 2023).

> **Prompt:** Now I will give you one category name. Please create a list which contains 10 visually similar categories of the provided category.
> For example: If I give you a category name: Acacia cochliacantha. You should return: [Acacia cambagei, Acacia calamifolia, Acacia campylacantha, Acacia cardiophylla, Acacia colei, Acacia colletioides, Acacia compacta, Acacia corymbosa, Acacia crocophylla, Acacia cuthbertii]
> Now, I give you this category name: Abaeis Nicippe.
> You should return:
>
> **Response:** [Eurema ada, Eurema alitha, Eurema andersonii, Eurema beatrix, Eurema blanda, Eurema brigitta, Eurema candida, Eurema celebensis, Eurema desjardinsii, Eurema esakii]

Table 7 shows examples of searched category names for each query class on three benchmark datasets. The results show that LLM can provide satisfactory responses using our prompts. After removing duplicates, we obtain 8913, 2318, and 99192 class names for ImageNet-LT (Liu et al., 2019), Place-LT (Zhou et al., 2017), and iNat18 (Van Horn et al., 2018) datasets, respectively. Then we search images for each queried name through the web (*e.g.*, Google/Duckduckgo Image Search Engine). After removing the dissimilar images, concretely, we collect 4.1M, 1.1M, and 3.6M images in 5012, 1895, and 20380 categories as auxiliary data. Fig. 7 shows the distribution of instance numbers for three datasets in each training epoch. It can be observed that 'Tail' is extended by auxiliary data for each dataset.

## C  ADDITIONAL EXPERIMENTS

### C.1  COMPARISON TO IMPROVED SOTA WITH DINOV2

As shown in Table 8, we re-implement LiVT (Xu et al., 2023) on DINOv2 (Maxime et al., 2023), which is the first work to apply ViT (Dosovitskiy et al., 2021a) to long-tail learning and leads the performance under the training from scratch paradigm. Our implementation differs only in that LVIT conducts MAE (He et al., 2022) training on the downstream dataset because training directly

**Table 8: Re-implementation of previous method with DINOv2.** We report the performance on three standard benchmark datasets (*i.e.*, ImageNet-LT, iNaturalist 18, and Place-LT).

| Methods | Backbone | Overall | Many | Medium | Few |
|---|---|---|---|---|---|
| **Results on ImageNet-LT with DINOv2 pretraining** | | | | | |
| LiVT(Bal-BCE) (Xu et al., 2023) | ViT-B | 79.4 | **84.9** | 78.2 | 68.5 |
| LiVT(Bal-CE) (Xu et al., 2023) | ViT-B | 79.6 | 84.3 | 78.3 | 71.1 |
| *Ours* | ViT-B | **82.0** | 84.7 | **81.5** | **76.2** |
| **Results on iNat18 with DINOv2 pretraining** | | | | | |
| LiVT(Bal-BCE) (Xu et al., 2023) | ViT-B | 84.5 | 84.4 | 85.4 | 83.3 |
| LiVT(Bal-CE) (Xu et al., 2023) | ViT-B | 85.0 | 85.7 | 86.2 | 84.2 |
| *Ours* | ViT-B | **87.0** | **86.4** | **87.4** | **86.7** |
| **Results on Place-LT with DINOv2 pretraining** | | | | | |
| LiVT(Bal-BCE) (Xu et al., 2023) | ViT-B | 49.6 | **52.4** | 49.7 | 45.2 |
| LiVT(Bal-CE) (Xu et al., 2023) | ViT-B | 49.5 | 49.2 | 51.3 | 46.1 |
| *Ours* | ViT-B | **50.8** | 49.4 | **52.4** | **49.2** |

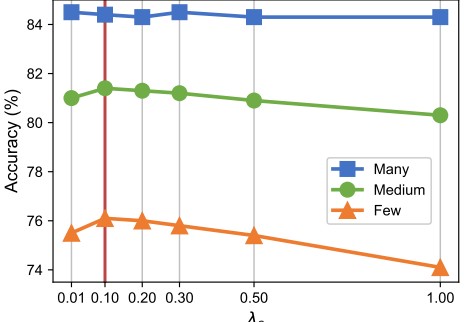

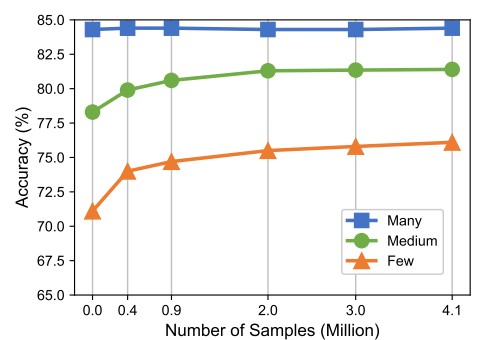

**(a) Ablation study on $\lambda_s$ in the proposed neighbor-silencing loss.**

**(b) Ablation study on the number of samples in the auxiliary dataset.**

**Figure 8:** More ablation studies. Experiments are conducted on ImageNet-LT (Liu et al., 2019).

on a long-tail dataset with randomly initialized parameters is difficult to converge, whereas we initialize directly with the weight from DINOv2. LiVT leverages the Bal-BCE (Xu et al., 2023) loss by default. We also implement Bal-CE (Xu et al., 2023)) to train LiVT with DINOv2. Table 8 demonstrates that our method shows superior performance on "Medium" and "Few" splits across three standard benchmarks. For example, our method surpasses LiVT(Bal-BCE) 3.2% and 7.6% on "Medium" and 'Few' in ImageNet-LT. Note that we set LiVT (Bal-CE) as the baseline method under three pre-training paradigms (training from scratch, CLIP, and DINOv2).

## C.2 EXTENDED ABLATION STUDY

**Effect of $\lambda_s$.** As shown in Fig. 9c, we study the effect of $\lambda_s$ in the proposed neighbor-silencing loss. The optional values are $\{0.01, 0.10, 0.20, 0.30, 0.50, 1.00\}$. It can be seen that as $\lambda_s$ increases to 0.1, the performance improves. However, when $\lambda_s$ increases to 1.0, the performance drops. This can be attributed that as $\lambda_s$ gradually increases, the proposed neighbor-silencing loss will gradually downgrade to the standard cross-entropy loss. In this case, the downstream dataset and the auxiliary dataset are treated equally during the training optimization, and the inconsistency between the network's optimization objective and the testing process leads to a decline in performance.

**Number of Auxiliary Samples.** As shown in Fig. 8b, we study the effect of the number of samples in the auxiliary dataset. We find that as the number increases from 0 to 0.9 million, there is a dramatic improvement in the accuracy in the few and medium categories, and relatively satisfactory performance is achieved, where +3.7% and 2.3% improvement in the few and medium categories,

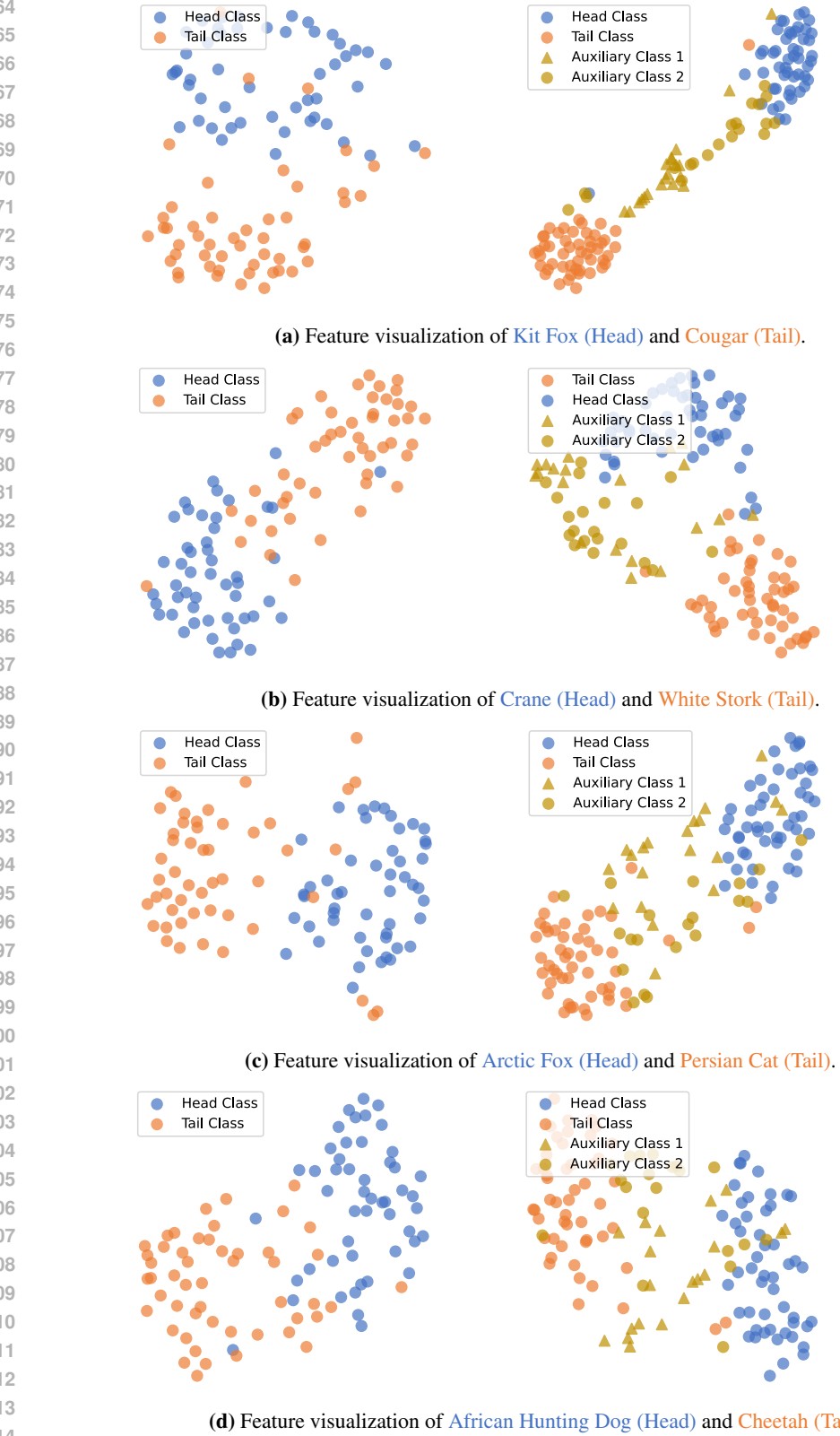

(a) Feature visualization of Kit Fox (Head) and Cougar (Tail).

(b) Feature visualization of Crane (Head) and White Stork (Tail).

(c) Feature visualization of Arctic Fox (Head) and Persian Cat (Tail).

(d) Feature visualization of African Hunting Dog (Head) and Cheetah (Tail).

Figure 9: **Feature visualization of confusing head and tail classes by UMAP (McInnes et al., 2020) on ImageNet-LT (Liu et al., 2019).** The left column shows the feature extracted by the model without auxiliary data, and the right is with the auxiliary fine-grained categories.

respectively. From 0.9 million to 4.1 million, the performance gradually increases. This indicates the data efficiency of our method.

### C.3   FEATURE VISUALIZATION

In Fig. 9, we provide more examples to demonstrate the effectiveness of auxiliary fine-grained categories on the feature separation for the head and tail classes. We conduct the experiments on ImageNet-LT (Liu et al., 2019) and train the models from random initialization. The left column shows the feature extracted by the model without auxiliary data, and the right is with the auxiliary fine-grained categories. The results show that training with auxiliary fine-grained categories benefits better feature separation between original head and tail classes.

### C.4   LONG-TAIL IN iNATURALIST18

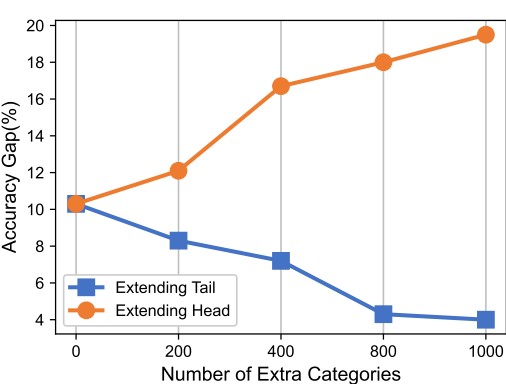

**Figure 10:** Effect of extending tail vs. extending head.

In Sec. 3.2, we validate the effect of granularity on the performance balance. Except for the granularity, we find that another difference between iNat18 and ImageNet-LT is that the number of tail categories in iNat18 is significantly larger than the number of head categories. To validate the effect of the proportion of tail categories, we sample 500 classes from the dataset pool, comprising 60 superclasses, with an imbalance ratio of 0.01. We conduct two sets of experiments: in the first set, we add extra categories to head classes (each category with more than 100 samples); in the second set, the extra categories are added to tail (each category with less than 20 samples). In both sets, the extra categories are fine-grained categories related to the original tail categories. As shown in Fig. 10, the results show that the long-tail benefits the performance balance, while the long-tail will exaggerate the imbalanced performance. This also validates our motivation of extending tail categories with fine-grained categories to balance the feature learning.

## D   DISCUSSIONS

### D.1   CONTRIBUTIONS

We summarize and discuss our main contributions as follows:

1) **A new perspective for long-tail learning from neighbor categories.** We investigate how to enhance long-tailed learning from open-set data, which is an understudied problem. Our pilot study (Sec. 3) highlights the granularity matters in long-tail learning (Sec. 3.2) and the need for auxiliary categories to improve generalization (Sec. 3.3). As shown in Fig. 2(c), traditional reweighting methods fail to generalize well. However, based on our finding in Sec. 3.2 that increased granularity of training data benefits long-tail learning ((Fig. 3)), we apply auxiliary fine-grained categories, which leads to better separation of the target classes (Fig. 2(d)). We also conduct studies on how to select auxiliary categories: inappropriate auxiliary data can even hinder long-tail learning (Fig. 4), and there exists a trade-off between the similarity and diversity of auxiliary data (Sec. 3.3). We believe these insights are valuable to the community.

2) **Fully automated data acquisition.** Inspired by our findings, we develop a fully automated pipeline for auxiliary data acquisition. As detailed in Sec. 4.1, we utilize GPT-4 API to query neighbor categories for target classes. Then, we retrieve images from the Web and automatically filter these images. We will release all the associated code.

3) **A new balanced loss with neighbor silencing.** As shown in Sec. 4.2, we design a new balanced loss with neighbor silencing for improving long-tailed learning with auxiliary data, which mitigates

the distraction of extra classes during training. After training, we directly mask out the classifier weights of auxiliary categories to obtain the final classifier. We find that this strategy works better than retraining a new one by linear probing.

## D.2 LIMITATIONS

This paper proposes to balance feature learning on downstream long-tail datasets by using visually similar categories. While it has achieved decent performance, there are still the following limitations. First, we use LLM (OpenAI, 2023) to obtain the names of similar categories. This step depends on the capability of the large language model; if the model has not seen or is unfamiliar with our query, then this step will fail. Second, we obtain images through the web, but we find that some categories are difficult to obtain online, such as those related to the iNat18 categories. For some special categories, we may need to look for more specialized websites to crawl data.

## D.3 FUTURE WORK

In future research, we consider collecting large-scale unlabeled data as an auxiliary dataset for downstream long-tail datasets and then using this dataset to balance feature learning. Since it is an unlabeled dataset, we can only consider its similarity to the downstream dataset, so compared to the data collection method in this paper, we can have feature learning on a larger scale. Secondly, we find that in a long-tailed distribution dataset, the distribution of superclasses also shows a long-tailed distribution in some datasets (*e.g.*, iNat18 (Van Horn et al., 2018)), we will also take into account the long-tail distribution of superclasses to achieve a better balance in feature learning.

