# OpenReview forum: "Granularity Matters in Long-Tail Learning"
_ICLR.cc/2025/Conference — ICLR 2025 Conference Withdrawn Submission_

### Official Review · Reviewer_U6mt · 2024-10-31

**Soundness:** 2
**Presentation:** 3
**Contribution:** 2
**Rating:** 6
**Confidence:** 3

**Summary:**

The paper proposes a new method for addressing long-tail learning problem via incorporating open-set data that belong to neighbor categories. The method is based on an interesting observation that datasets with finer granularity suffer less from the class imbalance issue. Authors of the paper first conduct a controlled experiment to support their observation, then describe the pipeline for querying neighbor categories from GPT-4 and filtering the web-crawled data. A neighbor-silencing re-balancing loss is proposed to balance the training, and the inference time classifier is obtained via masking out all the weights corresponding to neighbor categories. Experiments are conducted on three class-imbalanced benchmarks and the proposed method generally outperforms various previous approaches by a large margin.

**Strengths:**

1. The authors provide a novel observation of the relation between dataset granularity and model long-tail performance which, although somewhat counter intuitive, is supported by a well-constructed controlled experiment.

2. Experiment results are strong, showing that the proposed method outperforms baselines by a large margin.

3. The paper is well organized and easy to follow.

**Weaknesses:**

1. The reviewer has several questions regarding the rationale of the method and the results of the experiments. They are not necessarily weaknesses, but the reviewer expects them to be addressed. Please refer to the question part.

2. The reviewer suggests that the figure 3 should also demonstrate the exact accuracy of the head and tail categories in addition to their relative accuracy gap. Because it is possible that the more fine-grained dataset is more difficult to learn, and thus the major reason for the smaller accuracy gap is the significant accuracy degradation in head categories, which is undesirable. The experiment should clarify this point.

3. There are details that need improvement:
- In section 2.1, notations $\theta_f$ and $\theta_w$ should be explained.

- There is a citation error in section 2.1, line 160.

- Appendix D.1 suggests that there is a missing section 3.3. In fact, the reviewer would like to see the discussion that should have appeared in section 3.3.

**Questions:**

1. The rationale of the method is unclear. Since it is feasible to craw data from the internet, why not directly augment the tail categories, or at least incorporate the tail categories into the querying list for the searching engine together with the neighbor categories?
2. Since the setting does not consider performance on auxiliary categories, an alternative method would be simply treating all the data from neighbor categories as the augmentations for the target category (i.e., they share the same label). In this way, the overwhelming problem naturally disappears and therefore no need to introduce the weight $\lambda_s$. The reviewer would like to see the performance comparison  against this variant.
3. Comparing the results in table 2 and table 3, the reviewer wonders the discrepancy between the two results based on CLIP pre-trained model. Why the results are different?
4. Does the baseline in ablation study (table 6) use auxiliary categories? If it does, the reviewer suppose that the results should maintain consistent with the results in table 3. (Overall accuracy should be 77.9 instead of 79.6).

---

### Official Review · Reviewer_DAEc · 2024-11-04

**Soundness:** 3
**Presentation:** 2
**Contribution:** 2
**Rating:** 3
**Confidence:** 4

**Summary:**

This paper presents a strategy for classification in an imbalanced-class setting, which relies on expanding the training dataset via searching the internet for examples of classes related to rare classes in the existing dataset. The paper proposes a method for downloading additional data for nearby classes and a strategy for training from this additional data. The paper reports results on  ImageNet-LT and iNaturalist, and shows that the proposed method can improve over state-of-the-art methods for long-tail recognition without using external data.

**Strengths:**

1.	The method for the paper is well described and easy to understand.
2.	The authors show that the method can be applied on top of a number of prior approaches, displaying the versatility of the method as an add-on to existing approaches.
3.	It’s nice to see that approach also helps over large-scale pre-trained models like CLIP, a common concern for long-tail methods.

**Weaknesses:**

1.	Generally, it appears to me that the key improvement from the method is from downloading additional data similar to the rare classes. This is a fairly different setting than most long-tail approaches, which attempt to tackle the problem of learning only from the data you have. Given this, it’s difficult to compare this to most prior methods reported in the paper, and I would expect more thorough baselines to make up for this.
a)	For example, the authors highlight Iscen et. al., 2023, as related work, but I would have appreciated a more thorough comparison to Iscen et. al., both in terms of their method of obtaining additional data, and in terms of experimental results.
b)	Additionally, I would have appreciated a baseline which simply downloads more images by searching for the rare class from the internet directly.
2.	I’m confused about the focus on granularity in the writing. It seems the real win here is from getting more training data – which is totally fine (with appropriate baselines). But the writing + experiments around granularity paint a different picture.
a)	For example, Table 1 aims to show that a finer granularity helps with dataset imbalance. But a confounding factor here is the dataset size, which the comparison doesn’t control for. Similarly, to my understanding, Figure 3 is more about the distribution of classes, and not necessarily about the granularity. I would buy granularity if the proposed method used the same data, but with more labels or more fine-grained labels. Otherwise, this appears to be a story about more data, not granularity.

**Questions:**

I'd like the authors to address my concerns in Weaknesses above around the baseline comparisons (to Iscen et al, and to a simpler baseline), and the narrative around granularity.

---

### Official Review · Reviewer_86cN · 2024-11-04

**Soundness:** 2
**Presentation:** 2
**Contribution:** 2
**Rating:** 3
**Confidence:** 2

**Summary:**

This paper finds that finer-granularity datasets perform better in the face of imbalance. Using this finding, the authors propose using LLMs to define additional classes and gathering additional web images for these classes to augment the dataset. The authors also introduce a neighbor-silencing loss to account for these additional classes. This new method outperforms the baseline across different model types (scratch, CLIP, DINOv2).

**Strengths:**

- Empirical initial experiments that motivate the solution for the bigger picture problem
- Simple solution that is potentially scalable
- Strong improvement over the baseline numbers for a variety of image models

**Weaknesses:**

The implementation details (4.2) were a little unclear so I didn't fully understand the experimental setup. This leads to the following concerns.

I'm unsure if the results in the experimental section are a fair comparison. The baseline seems to be trained on less data, so the gains cannot be completely attributed to the new method. Similar concerns can be raised for the comparison with other methods. Regarding the fair comparison paragraph in 4.3, these methods were not designed for auxiliary classes so maybe you would need to collect additional images for the long-tail classes for the other methods and show that your method does better even though you are allocating resources towards auxiliary classes instead of the actual long-tail class.

It also seems like the new dataset has a higher proportion of long-tail (or long-tail neighbor) images. This changes the distribution of your training set. It would be good to make clear what part of the improvements are from using the auxiliary classes versus this change in distribution.

I would be happy to adjust my rating once these details are clarified or addressed.

**Questions:**

What do you do with existing images in the original dataset of auxiliary classes? Sometimes images can also belong to multiple classes. Would improving your pipeline with respect to these issues improve downstream performance?

Long-tail classes are often rare for a reason. What if it is difficult to find some of their auxiliary classes on the internet?

---

### Official Review · Reviewer_2XU9 · 2024-11-06

**Soundness:** 3
**Presentation:** 3
**Contribution:** 2
**Rating:** 5
**Confidence:** 5

**Summary:**

This paper addresses the challenge of training deep learning models on long-tail data distributions, where limited sample diversity hinders the learning of robust features, especially for tail classes. The authors propose a novel approach to overcome this issue by increasing dataset granularity. They introduce open-set auxiliary classes, which are visually similar to existing classes, to enhance feature representation learning for both head and tail classes. To generate these auxiliary categories, the method leverages large language models (LLMs) to search for and retrieve relevant images. Additionally, a neighbor-silencing loss is introduced to prevent auxiliary classes from disrupting the model’s focus on class discrimination within the target dataset. The approach is validated through extensive experiments on long-tail benchmarks, demonstrating superior performance over existing methods.

**Strengths:**

1. This work approaches long-tail learning from a novel angle by proposing to increase dataset granularity as a way to address data imbalance, differing from traditional re-sampling or re-weighting methods. The authors observe that datasets with finer granularity are typically less affected by imbalance, leading them to introduce category extrapolation to enhance granularity.

2. On several standard long-tail benchmarks, the proposed method outperforms other strong baselines using the same data volume, showcasing its effectiveness in the long-tail learning setting. Comprehensive ablation studies further demonstrate the impact of each component of the approach.

3. The paper is well organized and clearly written, making it easy for readers to understand the motivation, methodology, and experimental results.

**Weaknesses:**

1.Although the proposed method shows performance improvements, its resource costs, dataset construction time, and human resource costs are significantly higher. For example, the introduced data volume is nearly ten times that of the original dataset, indicating that researchers applying this method to other domains or datasets would need to invest substantial time and resources, making the method difficult for the community to adopt.

2.The authors expand the dataset by adding multiple times more data to introduce auxiliary categories. However, if these resources were instead directed toward expanding existing categories, could similar performance gains be achieved with less data? Discussing this would help the community and readers better understand the advantages and limitations of the method.

3.The paper lacks discussion of using large language models (LLMs) for addressing the long-tail issue in this field, as seen in methods like LTGC[1]. Additionally, exploring generative models as an alternative approach for auxiliary category expansion could offer promising solutions?

4.The work lacks data visualizations for the added categories, which limits the demonstration of the effectiveness of the proposed pipeline.

[1]Zhao Q, Dai Y, Li H, et al. LTGC: Long-tail Recognition via Leveraging LLMs-driven Generated Content[C]//Proceedings of the IEEE/CVF Conference on Computer Vision and Pattern Recognition. 2024: 19510-19520.

**Questions:**

1.In Table 2, why are the results on the iNaturalist18 dataset for the baseline and the proposed method trained from scratch higher than those based on CLIP?

2.What are the specific values for the neighbor category ratios across head, medium, and tail classes for different datasets?

3.Could you provide more visualizations of the additional categories across different datasets? I’m particularly curious if enough additional categories are generated for fine-grained datasets like iNaturalist. For the semantic level, the authors designed a DINO-based method, but does the web-crawling engine also introduce noisy labels?

4.In the limitations section, it mentions that “if the model has not seen or is unfamiliar with our query, this step will fail.” What is the failure rate across different datasets, and which categories tend to fail?

5.Compared to web-crawling for images, have you considered using generative models like Stable Diffusion 3.0 for image generation?

6.After expanding the categories, 4.1M, 1.1M, and 3.6M data points were collected. What image resolutions were used for training, and what was the training time on a 4-card RTX 3090 setup?

7.What value was set for the $\gamma_1$ and $\gamma_2$ threshold? Does it vary across different datasets?

8.How does filtering the data using DINOv2, rather than CLIP, affect models initialized in different ways?

9.For iNaturalist18, how many images are approximately generated per auxiliary category? If a generated category name is the same as an existing category in the dataset or is a synonym, does it impact the results?

10.Is the data open source?

11.Typo: DIONOv2 should be corrected to DINOv2.

---

### Author Response · Authors · 2024-11-15

We sincerely appreciate the reviewers for dedicating their time and effort to reviewing our work and for recognizing the potential contributions we may have made. The insightful feedback, constructive comments, and suggestions provided by the reviewers have significantly enhanced the quality and clarity of our work. We will incorporate these valuable suggestions into the revised version to further improve the content.

---

### Note · Authors · 2024-11-15

I have read and agree with the venue's withdrawal policy on behalf of myself and my co-authors.